# Secondary Genetic Events and Their Relationship to *TP53* Mutation in Mantle Cell Lymphoma: A Sub-Study from the FIL_MANTLE-FIRST BIO on Behalf of Fondazione Italiana Linfomi (FIL)

**DOI:** 10.3390/cancers17244027

**Published:** 2025-12-17

**Authors:** Maria Elena Carazzolo, Francesca Maria Quaglia, Antonino Aparo, Alessia Moioli, Alice Parisi, Riccardo Moia, Francesco Piazza, Alessandro Re, Maria Chiara Tisi, Luca Nassi, Pietro Bulian, Alessia Castellino, Vittorio Ruggero Zilioli, Piero Maria Stefani, Alberto Fabbri, Elisa Lucchini, Annalisa Arcari, Luisa Lorenzi, Barbara Famengo, Maurilio Ponzoni, Angela Ferrari, Simone Ragaini, Jacopo Olivieri, Vittoria Salaorni, Simona Gambino, Marilisa Galasso, Maria Teresa Scupoli, Carlo Visco

**Affiliations:** 1Department of Medicine, University of Verona, 37134 Verona, Italy; 2Research Center LURM (Interdepartmental Laboratory of Medical Research), University of Verona, 37134 Verona, Italy; antonino.aparo@univr.it (A.A.); mariateresa.scupoli@univr.it (M.T.S.); 3UOC di Ematologia, DAI Medico Generale, AOUI Verona, 37134 Verona, Italy; 4Department of Pathology and Diagnostics, University Hospital of Verona, 37126 Verona, Italy; 5Division of Hematology, Department of Translational Medicine, Università del Piemonte Orientale and AOU Maggiore della Carità di Novara, 28100 Novara, Italy; 6Department of Medicine, University of Padova and Unit of Hematology, Azienda Ospedale Università Padova, 35128 Padova, Italy; 7UOC Ematologia, Spedali Civili di Brescia, 25123 Brescia, Italy; alessandro.re@asst-spedalicivili.it; 8Hematology Unit, San Bortolo Hospital, 36100 Vicenza, Italy; 9Department of Hematology, Careggi Hospital and University of Florence, 50134 Florence, Italy; 10Clinical and Experimental Onco-Hematology Unit, Centro di Riferimento Oncologico di Aviano, Istituto di Ricovero e Cura a Carattere Scientifico, 33081 Aviano, Italy; 11Hematology Unit, AO Santa Croce e Carle, 12100 Cuneo, Italy; castellino.ale@ospedale.cuneo.it; 12Division of Hematology, ASST Grande Ospedale Metropolitano Niguarda, 20162 Milan, Italy; 13Struttura Complessa di Ematologia, Dipartimento Strutturale di Medicina Interna, Presidio Ospedaliero di Treviso, 31100 Treviso, Italy; pieromaria.stefani@aulss2.veneto.it; 14UOC Ematologia, Azienda Ospedaliero-Universitaria Senese, 53100 Siena, Italy; 15UCO Ematologia, Azienda Sanitaria Universitaria Giuliano Isontina, 34148 Trieste, Italy; elisa.lucchini@asugi.sanita.fvg.it; 16Unità Operativa Complessa di Ematologia, Azienda USL di Piacenza, Ospedale Guglielmo da Saliceto, 29121 Piacenza, Italy; 17Department of Histopathology, University of Brescia, ASST Spedali Civili di Brescia, 25123 Brescia, Italy; 18Pathology Department, San Bortolo Hospital, 36100 Vicenza, Italy; 19Faculty of Medicine and Surgery, Vita-Salute San Raffaele University and Pathology Unit, San Raffaele Hospital Scientific Institute, 20132 Milan, Italy; 20Hematology, Azienda USL-IRCCS di Reggio Emilia, 42123 Reggio Emilia, Italy; 21Division of Hematology, Department of Molecular Biotechnology and Health Sciences, University of Torino, 10126 Turin, Italy; 22Hematology Clinic, ASUFC Udine, 33100 Udine, Italy; 23Department of Diagnostic and Public Health, University of Verona, 37134 Verona, Italy; 24Department of Engineering for Innovation Medicine, Section of Biomedicine, University of Verona, 37134 Verona, Italy

**Keywords:** mantle cell lymphoma, POD24, MCL, *TP53*, Del 9p21.3 (*CDKN2A*), CNV, prognosis, NGS

## Abstract

Mantle Cell Lymphoma (MCL) is an aggressive non-Hodgkin lymphoma characterized by a highly variable clinical course and marked molecular heterogeneity. The analysis of targeted sequencing revealed the presence of secondary genomic events, such as copy number variations (CNVs), that may stratify patients according to their risk. CNV-based clustering may therefore represent a novel approach for identifying high-risk subgroups. Furthermore, the prognostic and biological relevance of *TP53* in disease progression is confirmed, and its integration with the CNV-defined clusters provides additional stratification of the cohort.

## 1. Introduction

Mantle Cell Lymphoma (MCL) is a rare and aggressive B-cell lymphoma, accounting for 4–7% of non-Hodgkin lymphomas (NHL). It is characterized by a continuous pattern of relapse and presents a highly heterogeneous clinical course [1,2]. Together with risk factors such as blastoid or pleomorphic morphology [3,4], the Mantle Cell Lymphoma International Prognostic Index (MIPI) [5], the nuclear proliferation index Ki-67 [6], and *TP53* aberrations [7,8,9,10], the time to first relapse or progression (POD) within two years of diagnosis has been independently associated with a higher risk of death [11].

Although these indices provide valuable prognostic information, they have not yet enabled the implementation of personalized therapeutic approaches in MCL. The integration of multi-omics approaches into diagnostic procedures may facilitate the detection of secondary events related to both treatment resistance and progression of the disease [12,13]. In particular, a group of broadly recognized genes with strong prognostic implications has been described, such as *ATM*, *TP53*, *NOTCH1/2*, *KMT2D*, *CCND1*, and *HNRNPH1* [7,14,15,16]. Specifically, *TP53* mutation, which occurs in 5–20% of cases [1,17,18], is the most impactful negative prognostic marker in patients with MCL, regardless of age, MIPI, delivered therapy, or morphology [17,19]. Nevertheless, MCL is also characterized by secondary genomic events such as copy number variations (CNVs), for instance, deletions (Del) at 17p, 13q33-q34, 1p22, 11q22-q23, 6q, 13q14 (*RB1*), 9p21 (*CDKN2A/B*), 9q22q31, and 10p15-p13, and amplifications (Amp) at 3q25-q29, 18q21-q22 (*BCL2*), and 12q13 (*CDK4*) [20].

CNVs have been less described in MCL than gene mutations. Among them, Del 9p21 (*CDKN2A*) has already been described as being associated with an adverse prognosis, independently of *TP53* mutation [10,21,22,23]. To evaluate the prognostic impact of CNVs in relation to *TP53*, we uniformly characterized, by targeted sequencing, a substantial cohort of patients with newly diagnosed MCL.

## 2. Materials and Methods

### 2.1. Study Design and Inclusion Criteria

The MANTLE-FIRST BIO (NCT04882475) is a study involving 16 major Italian institutions affiliated with the Fondazione Italiana Linfomi (FIL). To be included, patients had to satisfy the following criteria: (i) have available tissue for pathology revision and molecular studies; (ii) be diagnosed with MCL between 1 January 2008 and 30 June 2020; (iii) have been treated with disease management intent for symptomatic disease (excluding indolent or watch-and-wait approaches) using upfront therapy including rituximab; (iv) be relapsed or refractory (R/R) to induction therapy; and (v) 18–80 years old.

Patients were classified into early or late POD, considering a threshold of 24 months between the date of diagnosis and first relapse or progression [24]. Specifically, patients with early POD experienced relapse within 24 months, whereas those with late POD relapsed beyond 24 months. All tumor samples, preserved by means of Formalin-Fixed Paraffin-Embedded (FFPE) (Figure 1), were centralized at the Interdepartmental Laboratory of Medical Research (LURM) and the Department of Pathology of the University of Verona. The patient’s recruitment and the study were approved by an Independent Scientific Committee (IEC) in accordance with the Declaration of Helsinki, and all patients signed an informed consent form to be included in this research study, which represented a pre-specified sub-study of the MANTLE-FIRST BIO.

### 2.2. Extraction and Quality Control of Genomic DNA (gDNA)

Sections of 4 µm in size of formalin-fixed paraffin-embedded (FFPE) lymph nodes or extra-nodal sites of MCL naïve diagnostic patients were used for genomic DNA (gDNA) extraction through QIAamp DNA FFPE Advanced UNG Kit (Qiagen, Milan, Italy) [25]. Quality control was performed using Nanodrop (Life Real Biotechnology, Hangzhou, China), Qubit 4 Fluorometer (Thermo Fisher, Milan, Italy), and a fragment analyzer (Agilent Technologies, Milan, Italy). The minimal quantity of gDNA required was 0.5 μg with a concentration of at least 20 μg/uL.

### 2.3. Targeted Next Generation Sequencing (tNGS)

Targeted Next Generation Sequencing (tNGS) was performed on the Illumina platform 150 PE, adopting a customized panel comprising 37 genes: *ATM*, *BAX*, *BCL2*, *BCL6*, *BIRC3*, *BTK*, *CARD11*, *CCND1*, *CD36*, *CDKN2A*, *CREBBP*, *CXCR4*, *EP300*, *EWSR1*, *EZH2*, *HIST1H1E (H1-4)*, *HNRNPH1*, *KMT2D*, *MEF2B*, *MTOR*, *MYC*, *MYD88*, *NFKBIE*, *NOTCH1*, *NOTCH2*, *PLCG2*, *PTEN*, *RB1*, *RELA*, *SAMDH1*, *SETD2*, *SMARCA4*, *STAT3*, *TP53*, *TRAF3*, *UBR5*, and *NSD2.* The library preparation, sequencing, and variant calling were performed by an external service: Personal Genomics (Verona, Italy) [26]. The read lengths were 2 × 150 bp, obtaining 500 Mb as a median value for each sample. Copy number variations (CNVs) were annotated using CNVkit v.0.9.9 [27]. The CNVs were reported based on the coverage of the target region and their frequency in our cohort. In particular, the CNVs considered uncertain or with a frequency < 6 in the cohort were excluded from the statistical analysis. Instead, the *TP53* mutation was first automatically annotated with MuTec2 v4.1, referring to the genome Hg38, and secondly checked manually through the Integrative Genomics Viewer (IGV) software (Version IGV_2.17.1) [28]. In total, 11 patients (8 early POD and 3 late POD) presented *TP53* mutation with a variant allele frequency (VAF) > 10%, while 10 mutations were assessed as subclonal, considering the VAF between 2 and 6; the *TP53* mutation was distributed across exons 4 to 11. The two exons prevalently hit are exons 5 and 8, with a recurrence of 5 mutations each.

### 2.4. Statistical Analysis

The statistical analyses, performed in R v4.1.2, adopted the endpoint of time to first relapse or progression of disease (POD), defined as the interval between diagnosis and the first occurrence of relapse or refractoriness after frontline therapy. CNVs and clinical features were visualized by oncoprints. Group comparisons of categorical variables were performed using Fisher’s exact test. Pairwise association analyses were performed: among CNVs using the Phi coefficient with *p*-values from Fisher’s exact test; and between CNVs (encoded as present = 1/absent = 0) and clinical variables using: Phi for binary–binary (*TP53* status, sex, age group <65/≥65, Ki-67 <30/≥30, POD, and SOX11), Cramer’s V for binary–multicategory (MIPI and morphology), and point-biserial correlation for binary–continuous where available. Only associations with *p* < 0.05 are shown. Unless otherwise specified, a two-sided *p*-value ≤ 0.05 or FDR-adjusted *p* ≤ 0.05 was considered statistically significant. Survival curves were estimated using the Kaplan–Meier method and compared using the log-rank test. Univariate and multivariate Cox proportional hazards models [29,30] were applied to identify prognostic factors, and results were reported as hazard ratios (HRs) with 95% confidence intervals (CIs). Variables significant (*p* ≤ 0.05) in univariate analyses were included in the multivariate model [3,31], which was built using a backward stepwise selection [32] procedure. For unsupervised clustering of CNV profiles, we applied non-negative matrix factorization (NMF) using binary data (presence/absence). Clusters showing similar Kaplan–Meier curves were further collapsed, resulting in two molecular groups. These clusters were subsequently correlated with clinical and molecular variables, and survival analyses (Kaplan–Meier, log-rank, and Cox regression) were performed. Prognostic effects of CNV clusters were also visualized using forest plots.

## 3. Results

### 3.1. Clinical Features of the Analyzed Cohort

The 73 MCL patients had a median age of 65 years (range, 40–80), calculated at the time of initial diagnosis, and 75% were male. Clinical and pathological characteristics of the 73 included patients, divided according to time to POD, are listed in Table 1. The median time to POD of the entire cohort was 35 months (range, 3–150 months). When comparing clinical and pathological variables between patients with early POD (*n* = 27, 37%) or late POD (*n* = 46, 63%), we found that MIPI (low-intermediate vs. high, *p* = 0.03), tumor morphology (*p* = 0.003), and *TP53* mutation (*p* = 0.001) were significantly differentially represented between the two groups (Table 1).

### 3.2. Copy Number Variations (CNVs) Identification and Description

A total of 273 CNV signatures were identified, which were subdivided into 18 types of alterations (11 amplifications and 7 deletions; Appendix A). The most recurrent CNVs were Amp 3q26-q28 (*BCL6*, 49%), followed by Amp 5q35.3 (*HNRNPH1*, 45%), Del13q14 (*RB1*, 34%), and Del 11q22.3-q23.2 (32%). Despite being usually more frequently described, 17p deletion (Del17p) [7,20] was found in only 3 of our patients, thus preventing statistical considerations for this cohort. Interestingly, Del 13q14 (*RB1*, *p* = 0.02, 52% early vs. 24% late), Del 6q (*p* = 0.01, 37% early vs. 11% late), and Del 9p21.3 (*CDKN2A*, *p* = 0.02, 33% early vs. 11% late) were significantly more prevalent in early POD patients as compared with late POD (Figure 2).

We next analyzed possible correlations between CNVs themselves and the relationship between their occurrence and clinical features (Appendix A). It emerged that Amp 7q11.2 (*CD36*), Amp 7p22.1 (*CARD11*), and Amp 7q26-q35; Del 9p21.3 (*CDKN2A*) and Del 9q22-q31; and Amp 3q26-q28 (*BCL6*) and Amp 3p21.1-p26 positively correlated with each other. Contrarily, Amp 3q26-q28 (*BCL6*) and Amp 18q21.3 (*BCL2*); Amp 5q35.3 (*HNRNPH1*) and Del 9p21.3 (*CDKN2A*) were negatively correlated.

### 3.3. Prognostic Factors in Terms of Time to POD: Clinical and Molecular Features

In univariate analysis, the variables significantly associated with a shorter time to POD were *TP53* mutation (*p* = 0.0013, HR = 2.3, Figure 3A), Del 9p21.3 (*CDKN2A*, *p* = 0.011, HR = 2.1, Figure 3B), high MIPI (vs. intermediate and low, *p* = 0.026, HR = 1.4; Figure 3C), and age ≥ 65 (*p* = 0.018, HR = 1.8; Figure 3D).

The Cox model showed that a high MIPI score (*p* = 0.009, HR = 2.43) and *TP53* mutation (*p* = 0.016, HR = 2.26) were independently associated with a shorter time to POD, indicating that the combined presence of these variables could effectively identify patients with a tendency toward a shorter time to POD.

### 3.4. Identification of Molecular Clusters That Provide Risk Stratification

In the attempt to improve risk stratification of CNVs, we used the NMF, which initially identified six different molecular clusters (Figure 4A). Post hoc survival analysis based on time to POD showed comparable Kaplan–Meier curves with similar patterns among certain groups. Thus, we merged the clusters with overlapping survival profiles, obtaining two distinct CNV-defined clusters. Cluster A (C_A_) combined original groups 1 and 6, whereas Cluster B (C_B_) encompassed groups 2–3–4–5 (Figure 4B).

C_A_ (12 early POD and 8 late POD) was associated with a significantly shorter time to POD than C_B_ (15 early POD and 38 late POD; HR = 0.50; *p* = 0.012; Figure 4C). Furthermore, the 20 patients who were identified as C_A_ were characterized by adverse clinicopathological features, with 4 (20%) patients having blastoid morphology, 13 (65%) having Ki67 ≥ 30, 10 (50%) having a high MIPI score, and 8 (40%) harboring *TP53* mutations. Instead, the cluster C_B_ was composed of 53 patients, of whom 2 (4%) presented with blastoid morphology, 21 (40%) with Ki-67 ≥ 30, 24 (45%) with a high-MIPI score, and 13 (25%) with *TP53* mutation. Surprisingly, and despite their more favorable clinical behavior, the C_B_ was enriched for Del 9p21.3 (*CDKN2A*), including 5 cases with co-occurring *TP53* mutation.

Then, we fitted a further multivariable Cox model, including the previously reported risk factors together with cluster assignment (Appendix A), which showed that C_A_ was significantly associated with a shorter time to POD compared with C_B_ (*p* = 0.007, HR = 0.37).

### 3.5. TP53 Mutation and Its Interaction with the Clustering Analysis

*TP53* mutation was detected in 28% of the patients (21 out of 73; Figure 3A) and was significantly associated with a shorter time to POD, regardless of VAF (2% to >10%). Stratification by *TP53* status highlighted differences in CNV distributions (Figure 5). Amp 6p22.1-p25.1 (*H1-4*, *p* = 0.02) was found to be significantly more prevalent in patients with *TP53* mutation compared to the wild type (WT).

We performed stratification of the clusters according to the presence of *TP53* mutation. Patients were divided into four groups with different clinical behavior according to cluster and *TP53* mutation (C_A_/*TP53*-WT, C_A_/*TP53*-mut, C_B_/*TP53*-WT, C_B_/*TP53*-mut; Figure 6). Interestingly, while patients with *TP53* mutation had a significantly inferior time to POD, regardless of the cluster they belonged to, patients with WT *TP53* could be divided into C_A_/*TP53*-mut vs. C_B_/*TP53*-WT *p* = 0.001, HR = 3.92; and C_B_/*TP53*-mut vs. C_B_/*TP53*-WT *p* = 0.014, HR = 2.23, two different groups showing a trend toward shorter time to POD (C_A_/*TP53*-WT vs. C_B/_*TP53*-WT *p* = 0.048; HR = 1.87), as shown in Figure 6. The C_B_/*TP53*-WT group exhibited the longest time to POD among all groups.

## 4. Discussion

In the present study, we described the interactions between CNVs and *TP53* mutations in 73 unselected patients with MCL who had experienced treatment failure, namely relapse or progression after standard induction immunochemotherapy. We confirmed that a high MIPI score, *TP53* mutations, and Del 9p21.3 (*CDKN2A*) were the strongest independent predictors of shorter time to POD. Furthermore, we identified two CNV-based clusters that could refine the prognostic predictivity of the *TP53* mutation itself, effectively increasing our ability to discriminate prognosis in patients with wild-type *TP53*.

Our results, while confirming the fundamental role of *TP53* annotations in MCL [8,33], which are well known to affect survival endpoints [33,34] and responses to treatments in MCL patients [35,36], shift attention to the need to reconsider the important role of CNVs in this disease. Overall, as widely described in the literature, MCL presents great genomic heterogeneity [7,34,37,38], as further highlighted by the extensive use of high-throughput technologies. Even though the relevance of secondary genomic signatures [12,20] has been less investigated, we pointed out how genomic instability could further explain both the MCL genomic heterogeneity and its impact on the progression of the disease.

Among single genetic alterations, the univariate analysis reported *TP53* mut (*p* = 0.001, HR = 2.3) and Del 9p21.3 (*CDKN2A*, *p* = 0.01, HR = 2.1) as the two molecular factors leading to significantly worse times to POD. Interestingly, 42% of patients presented a co-occurrence of these alterations, confirming previous literature [4,21,22,23,39,40]. To gain a deeper insight into the relevance of CNVs in our cohort, we applied an unbiased analysis to identify different molecular groups. We identified two different clusters, C_A_ and C_B_, which showed a diverse distribution of CNVs and were associated with significantly different time to POD (*p* = 0.012, HR = 0.50), with C_A_ effectively recognizing high-risk patients. C_B_ comprised patients harboring Del 9p21 (*CDKN2A*), who were prevalently associated with early POD. Moreover, the *TP53* hotspots identified in co-occurrence with the Del 9p21 (*CDKN2A*) were associated with a better prognosis compared to those identified in the C_A_. This aspect may account for the observed advantage of C_B_ with respect to disease progression.

To further confirm the significance of our unbiased analysis, we integrated the *TP53* mutation into the clustering model. The clusters enriched for *TP53* mutation predictivity, being associated with inferior time to POD (C_A_ and C_B_/*TP53*-mut vs. *TP53*-WT *p* = 0.001, HR = 3.92; and *p* = 0.014, HR = 2.23, respectively) as compared with the WT ones (Figure 6). Among *TP53*-WT clusters, we identified patients with significantly shorter time to POD (C_A_-*TP53*-WT), underlining that biological markers research can add to the recognition of *TP53* status. In addition, some preliminary data indicate that there might be variability in the prognostic impact of the *TP53* mutation itself, according to the affected hotspot, and the actual impact that the mutation may confer on gene function [9,41].

*CDKN2A* is a gene located in the p21 arm of chromosome 9, which is directly involved in the regulation of the cell cycle, acting as a tumor suppressor. Its deletion affects different pathways, including Rb1 and p53, leading to a dysregulation of the normal cell cycle [23]. The Del 9p21 (*CDKN2A*) is also linked to a multitude of pathways, such as the metabolic pathways that could further explain the possible involvement in the progression of the disease and drug resistance [42,43,44]. In our cohort, we identified co-occurrence of *TP53* mutation and Del 9p21 (*CDKN2A*) with a prevalence in cluster B (C_B_). Preliminary stratification analysis and correlations with clinical data (such as histopathological and first-line treatments) should associate the co-occurrence of these alterations with a poor prognosis and a tendency towards treatment resistance To this end, we aim to strengthen our results by conducting additional analyses, including in vitro experiments.

We acknowledge that our series, due to its retrospective nature, may have suffered from selection bias based on material availability in single centers. We must also consider that, due to the inclusion criteria of the study, which required patients who had already experienced relapse or progression, our series has likely been enriched in adverse genetic events, as compared with unselected series taken at the time of diagnosis. Finally, our analysis did not encompass comprehensive clinical and pathological evaluations, as well as somatic mutations in genes other than *TP53*, which may have contributed to the clinical behavior of the included patients. All these analyses will be the subject of the MANTLE-FIRST BIO study itself.

## 5. Conclusions

In conclusion, we confirmed the prognostic relevance of *TP53* mutation and Del 9p21 (*CDKN2A*) alterations in MCL patients. Both genetic signatures were enriched in patients with shorter time to first POD. Finally, the molecular clustering based on CNVs could discriminate patients with *TP53*-WT at a higher risk of early disease progression.

## Figures and Tables

**Figure 1 cancers-17-04027-f001:**
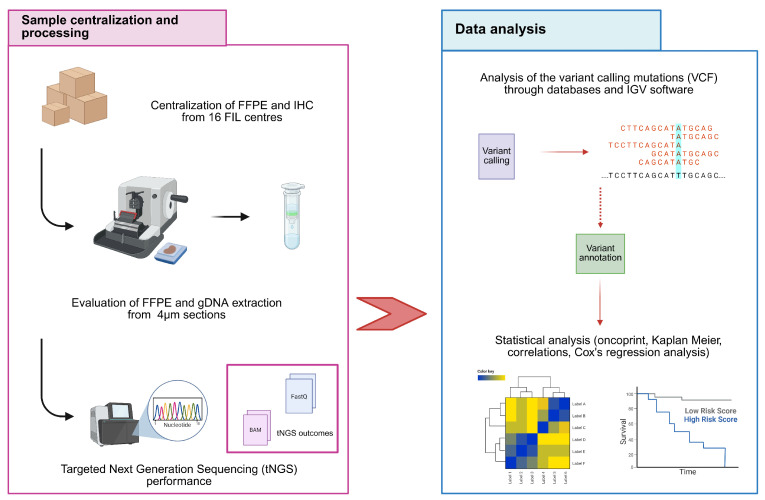
Graphical workflow: on the (**right**), the wet-lab steps (dissection of FFPE, gDNA extraction, and library preparation). On the (**left**), the bioinformatic tools adopted to perform data analysis (CNVkit, *TP53* annotation, and data analysis).

**Figure 2 cancers-17-04027-f002:**
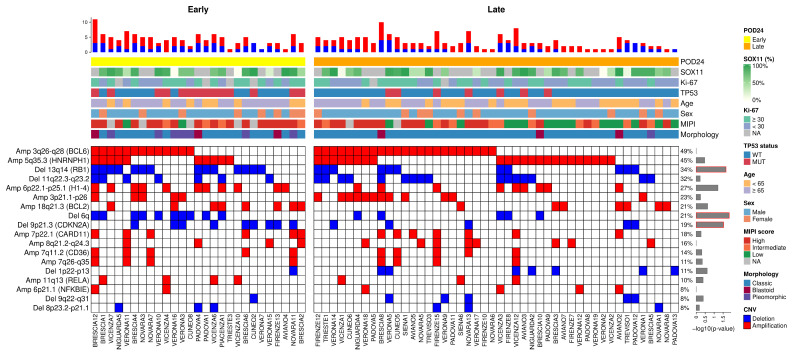
Recurrent CNVs were divided based on the classification of early POD (on the left) and late POD (on the right). The blue boxes are the deletions, while the red ones are the amplifications. On the top were reported: SOX11 (continuous value), Ki67 (≥30%, <30% and NA), *TP53* status, age (≥65 and <65), sex, MIPI score (high, intermediate, and low), and morphology (classic, blastoid, and pleomorphic). Each row reports the percentage and the *p*-value (at log10(*p*-value) scale, on the right) for each CNV reported. Del 13q14 (*RB1*), Del6q, and Del 9p21.3 (*CDKN2A*) showed a significant *p*-value (red box) when comparing early vs. late POD.

**Figure 3 cancers-17-04027-f003:**
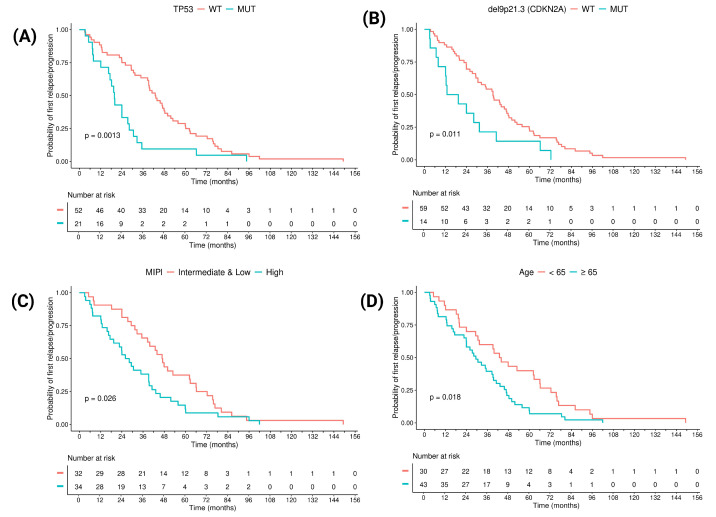
Kaplan–Meier plots of *TP53* mutation vs. the WT (**A**), Del9p21.3 (*CDKN2A*) mut vs. Del9p21.3 (*CDKN2A*) WT (**B**), MIPI high vs. MIPI intermediate and low (**C**), and age > 65 vs. <65 (**D**), regardless of the classification in early or late POD.

**Figure 4 cancers-17-04027-f004:**
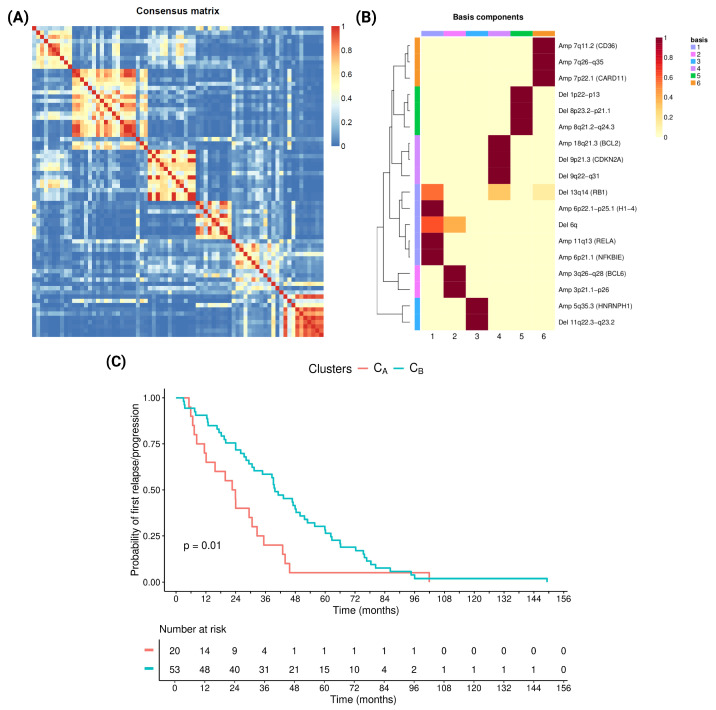
(**A**) Consensus matrix heatmap from NMF applied to binary CNV calls (presence/absence). (**B**) The cluster A (C_A_) is characterized by the deletions of 13q14 (*RB1*), 6q, and amplification of 6p22.1-p25.1 (*H1-4*), 11q13 (*RELA*), 6p21.1 (*NFKBIE*), 7p22.1 (*CARD11*), 7q11.2 (*CD36*), and 7q26-q35. Cluster B (C_B_) was characterized by the deletion of 11q22.3-q23.2, 9p21.3 (*CDKN2A*), 9q22-q31, 1p22-p13, 8p23.2-p21.1, and the amplifications of 3q26-q28 (*BCL6*), 3p21.1-p26, 5q35.3 (*HNRNPH1*), 18q21.3 (*BCL2*), and 8q21.2-q24.3. (**C**) Kaplan–Meier plot (with log rank *p*-value) of POD of C_A_ vs. C_B_, curve comparison: *p* = 0.01, HR = 0.50.

**Figure 5 cancers-17-04027-f005:**
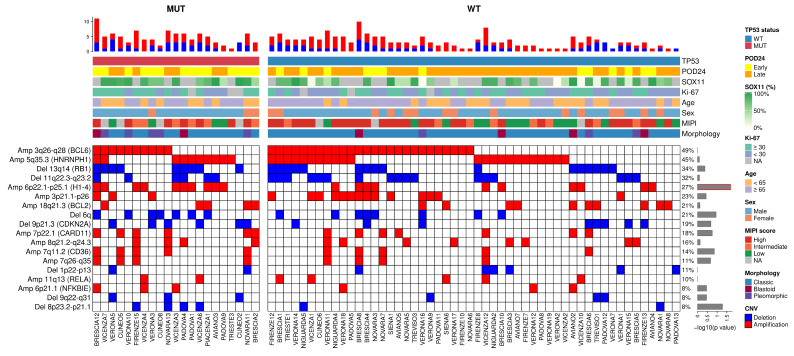
Recurrent CNVs stratified by *TP53* status (mut-red, WT-blue). Each row reports the frequency (percentage) and the *p*-value (Fisher’s exact test) for each CNV reported. Amp 6p22.1-p25.1 (*H1-4*) showed a significant *p*-value (red box) comparing *TP53* mut vs. WT.

**Figure 6 cancers-17-04027-f006:**
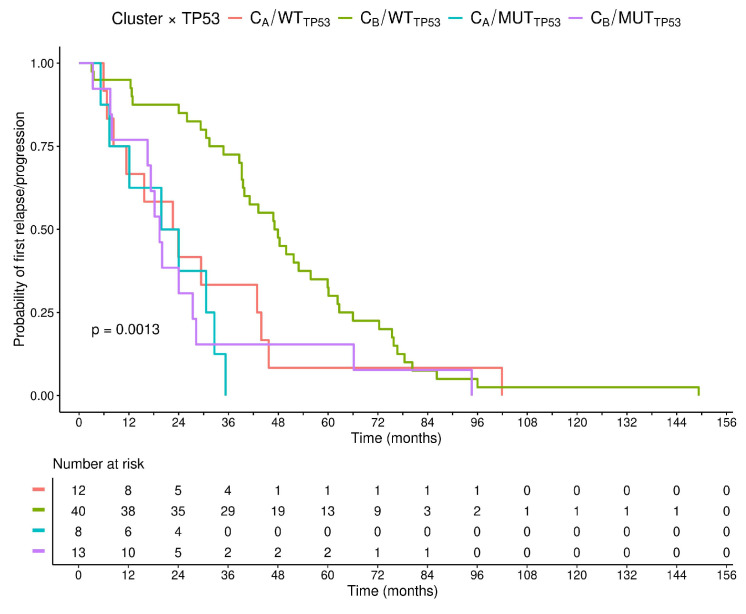
Kaplan–Meier plots associated with C_A_/*TP53*-WT (red) vs. C_A_/*TP53*-mut (blue) vs. C_B_/*TP53*-WT (green) vs. C_B_/*TP53*-mut (purple).

**Table 1 cancers-17-04027-t001:** Clinical characteristics of 73 newly diagnosed MCL patients. Patients were classified according to time to progression/relapse (POD) into early POD (≤24 months, *n* = 27) and late POD (>24 months, *n* = 46). For each variable, the distribution is reported for the whole cohort and for the early POD vs. late POD subgroups. Comparisons between early and late POD were performed using Fisher’s exact test, while for the median time to POD *, it was performed using the Wilcoxon test (*p* = 1.34 × 10^−12^).

Total (*n* = 73)		Early POD Tot (*n* = 27)	Late POD Tot (*n* = 46)	*p*-Value
**Gender**				
Male	55 (75%)	19 (70%)	36 (78%)	*p* = 0.57
Female	18 (25%)	8 (30%)	10 (22%)	
**Age**				
<65	30 (41%)	9 (33%)	21 (46%)	*p* = 0.33
≥65	43 (59%)	18 (67%)	25 (54%)	
**Alive**	18 (25%)	3 (11%)	15 (33%)	*p* = 0.05
**Dead**	55 (75%)	24 (89%)	31 (67%)	
**MIPI**				
Low	17 (23%)	2 (7%)	15 (33%)	
Intermediate	14 (19)	4 (15%)	10 (22%)	*p* = 0.03 *
High	34 (45%)	16 (59%)	18 (39%)	
NA	8 (11%)	5 (19%)	3 (6%)	
**Morphology**				
Blastoid	6 (8%)	3 (11%)	3 (7%)	*p* = 0.003 *
Classic	59 (81%)	17 (63%)	42 (91%)	
Pleomorphic	8 (11%)	7 (26%)	1 (2%)	
**Ki-67%**				
<30	33 (45%)	9 (33%)	24 (52%)	
≥30	34 (47%)	14 (52%)	20 (44%)	*p* = 0.30
NA	6 (8%)	4 (15%)	2 (4%)	
**Median time to POD 24**	34.8 months	12.7 months	47.5 months	NA
***TP53* mut**	21 (29%)	14 (52%)	7 (15%)	*p* = 0.001 *

## Data Availability

Data is contained within the article or Appendix A. Further, data that support the findings of this study are available from the corresponding author upon reasonable request.

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
