# Peer review of "Secondary Genetic Events and Their Relationship to *TP53* Mutation in Mantle Cell Lymphoma: A Sub-Study from the FIL_MANTLE-FIRST BIO on Behalf of Fondazione Italiana Linfomi (FIL)"

_cancers, 2025, doi:10.3390/cancers17244027_

Round 1
Reviewer 1 Report
Comments and Suggestions for Authors
The authors are commended on their methodical analysis of TP53 mutation and CNVs in a cohort of relapsed MCL patients, that recapitulates the significant of TP53 mutations and Del9p21 alterations in this cohort. The CNV clustering in TP53 WT is a novel predictor for early relapse and may find clinical significance in future.
Minor revisions suggest:
Please modify the statement in methods stating "treated with curative intent", given that mantle cell lymphoma is incurable and where frontline treatment is aimed at attaining durable remission.
Please be clearer in methods on how early verus late POD are defined (line 101). It is clear in the table 1 legend but should clearly be stated in text. Please clarify (line 115) that non-nodal tissue samples utilised were involved with MCL.
Please review the manuscript for minor typographical errors for correction, including in citations.
In discussion be further elaborate on the unexpected Cb enrichment for Del9p21.3 including co-occuring TP53 mutation.
Author Response
Thank you very much for taking the time to review our manuscript. We have carefully considered all your suggestions, and we have accepted them. The changes are highlighted in red in the revised manuscript for your convenience. We also improved the discussion regarding the cluster B and the involvement of del CDKN2A. We are pleased that the manuscript captured your interest in this topic, which we believe is highly relevant to current research in Mantle Cell Lymphoma.
Response 1: We modified the statement “treated with curative intent” with “standard induction therapy ”
Response 2: We added in line 101 the explanation around the definition of early-late POD, and in line 115 we added the expression extra-nodal sites
Response 3: We added a brief explanation in the discussion section (in red) addressing Cluster B and the possible reason for its lower aggressiveness compared with Cluster A.
Response 4: We checked and corrected the typos and references in the text
Reviewer 2 Report
Comments and Suggestions for Authors
This is a well-established, well-written manuscript that addresses a significant area of unmet clinical need in Mantle Cell Lymphoma (MCL): the refinement of prognostic stratification beyond standard indices. The authors successfully leverage a multi-center cohort from the Fondazione Italiana Linfomi (FIL) to investigate the interplay between TP53 mutations and secondary Copy Number Variations (CNVs).
The study is logically structured, the methodology is sound, and the conclusions are supported by the data presented. The manuscript makes a valuable contribution to the field by proposing a novel molecular clustering approach that improves risk assessment, particularly for patients with Wild-Type (WT) TP53.
Minor Points and Suggestions
- Figure 1 Typos: There is a typo in the flowchart in Figure 1. The box in the top left reads "Sample centralizatoin" instead of "Sample centralization". This should be corrected for the final publication.
- Resolution: Please ensure that the oncoprints (Figure 2 and Figure 5) are rendered at high resolution in the final proof, as the gene names and sample IDs can be dense.
- Cluster B Characteristics: It is noted that Cluster B was associated with better outcomes despite being enriched for CDKN2A deletions. The authors briefly mention this is surprising; expanding slightly on why this specific combination in Cluster B might be less aggressive (perhaps mutual exclusivity with other drivers not tested?) would add depth to the discussion.
Author Response
Thank you very much for taking the time to review our manuscript.
We have carefully considered all your suggestions, and we have accepted them. The changes are highlighted in red in the revised manuscript for your convenience. We also improved the discussion regarding cluster B and the involvement of del CDKN2A. We are pleased that the manuscript captured your interest in this topic, which we believe is highly relevant to current research in Mantle Cell Lymphoma.
Response 1: We correct the Figure 1 typos
Response 2: We uploaded the figures with a higher resolution
Response 3: We added a brief explanation in the discussion section (in red) addressing Cluster B and the possible reason for its lower aggressiveness compared with Cluster A.